# Modelling the Relationship between Match Outcome and Match Performances during the 2019 FIBA Basketball World Cup: A Quantile Regression Analysis

**DOI:** 10.3390/ijerph17165722

**Published:** 2020-08-07

**Authors:** Shaoliang Zhang, Miguel Ángel Gomez, Qing Yi, Rui Dong, Anthony Leicht, Alberto Lorenzo

**Affiliations:** 1Division of Sport Science & Physical Education, Tsinghua University, Beijing 100084, China; zslinef@mail.tsinghua.edu.cn; 2Facultad de Ciencias de la Actividad Física y del Deporte (INEF), Universidad Politécnica de Madrid, 28040 Madrid, Spain; miguelangel.gomez.ruano@upm.es (M.Á.G.); alberto.lorenzo@upm.es (A.L.); 3School of Physical Education and Sport Training, Shanghai University of Sport, Shanghai 200438, China; 4Shanghai Key Lab of Human Performance, Shanghai University of Sport, Shanghai 200438, China; 5China Basketball College, Beijing Sport University, Beijing 100084, China; rdong39@outlook.com; 6Sport and Exercise Science, James Cook University, Townsville, QLD 4810, Australia; Anthony.Leicht@jcu.edu.au

**Keywords:** team sport, elite athletes, basketball performance analysis, quantile regression

## Abstract

The FIBA Basketball World Cup is one of the most prominent sporting competitions for men’s basketball, with coaches interested in key performance indicators (KPIs) that give a better understanding of basketball competitions. The aims of the study were to (1) examine the relationship between match KPIs and outcome in elite men’s basketball; and (2) identify the most suitable analysis (multiple linear regression (MLR) vs. quantile regression (QR)) to model this relationship during the men’s basketball tournament. A total of 184 performance records from 92 games were selected and analyzed via MLR and QR, using 10th, 25th, 50th, 75th and 90th quantiles. Several offensive (Paint Score, Mid-Range Score, Three-Point Score, Offensive Rebounds and Turnovers) and defensive (Defensive Rebounds, Steals and Personal Fouls) KPIs were associated with match outcome. The QR model identified additional KPIs that influenced match outcome than the MLR model, with these being Mid-Range Score at the 10th quantile and Offensive Rebounds at the 90th quantile. In terms of contextual variables, the quality of opponent had no impact on match outcome across the entire range of quantiles. Our results highlight QR modelling as a potentially superior tool for performance analysts and coaches to design and monitor technical–tactical plans during match-play. Our study has identified the KPIs contributing to match success at the 2019 FIBA Basketball World Cup with QR modelling assisting with a more detailed performance analysis, to support coaches with the optimization of training and match-play styles.

## 1. Introduction

The FIBA Basketball World Cup is one of the most prominent basketball competitions in the world, with elite players from various countries competing against each other for the title of World Champion. This competition provides an opportunity for performance analysts and coaches to gain a deeper understanding of the modern technical and tactical behaviors undertaken by the world’s best basketball players and teams. However, very little research has been conducted at this elite level with examination of the relationships between some match behaviors and success; such research could potentially be valuable, in order to optimize the training and coaching processes.

Previously, technical behaviors or performances were demonstrated to have a crucial impact on match outcome [1]. Specifically, Defensive Rebounds were identified as the most common key performance between indicator/s (KPI/s) that differentiated winning and losing matches [2,3,4]. Additionally, a high success rate of fast breaks or transition play (i.e., rapid switch from defense to offence leading to a score) was identified as a distinguishing factor between winning and losing teams [5]. Ball coordination between inside and outside playing positions that enhance three-point shot opportunities was also acknowledged as important for winning matches [6]. Despite several studies investigating aspects of basketball technical behavior, the relationship between match outcome and KPIs has not been identified repeatedly, with empirical studies often reporting inconsistent findings. For example, Gómez et al. [3] reported that three-point field-goal achievements were a KPI for match success in the National Basketball Association (NBA), whereas Mikolajec et al. [7] indicated that there was no robust effect of three-point field-goals scored on match outcome. An important factor that has been rarely considered for this inconsistency is the various statistical techniques utilized by previous studies. For example, the relationship between match KPIs and outcome in most previous studies was considered as a linear one, with estimation by linear equations. However, the relationship between match KPI and outcome was suggested to be likely nonlinear [8]. As a result, the use of advanced statistical methods may be imperative for any examination of the effect of KPI on match outcomes. 

Consequently, this study addressed the following key questions: Does KPI influence final match outcome? If so, does this influence change with different levels of match outcome (i.e., point difference)? Considering the likely non-linear relationships [8] and need for appropriate analyses, quantile regression (QR) modelling was employed to tackle these questions [9]. The QR approach enables the examination of various conditional quantiles of the dependent variable, thereby revealing a range of heterogeneity in the analysis of the differences in the final scores [9]. Therefore, the QR methodology, originally developed by Koenker and Bassett [9,10], employs a more comprehensive and elaborate analysis, for a deeper understanding of the contributions of independent variables, by characterizing the entire conditional distribution of the dependent variable. In contrast, multiple linear regression (MLR) only describes the relationship between the mean conditional distribution of dependent variables and independent variables. Therefore, multiple linear regression (MLR), as a baseline model, and quantile regression (QR), as an advanced model, may enable a simultaneous exploration of the relationship between independent variables and dependent variables. Concurrent comparison would provide coaches and performance analysts with clarity and the importance of QR application within sports science for the future.

Based on the abovementioned, the aims of this study were to (1) examine the relationship between match KPI and outcome in elite men’s basketball and (2) identify the most suitable analysis (MLR or QR) to model this relationship. We hypothesized that the results from the QR model would provide more detailed explanations (i.e., non-linear performances) than the MLR model for match outcome.

## 2. Materials and Methods

### 2.1. Sample

This study was a retrospective analysis of publicly available data from the official FIBA Basketball website (http://www.fiba.basketball/basketballworldcup/2019/teamstats). The sample comprised 184 performance records from 92 games during the men’s basketball tournament at the 2019 FIBA Basketball World Cup. The variables examined included situational variables, offensive variables and defensive variables (Table 1) and were in accordance with those employed by Sampaio et al. [4,11]. In order to control for the situational variable effects, quality of the opponent was determined via a k-means cluster analysis [12]. This analysis put into groups the games of the sample in function of winning percentage. What this analysis does is assign a number of objects to a group (or cluster), so that the objects of the same cluster are more similar (in one way or another) among them than between the objects of another cluster [13]. The quality of the opponent was categorized into two groups: strong teams (winning = 68 ± 14%) and weak teams (winning = 24 ± 17%). Furthermore, normalization of all KPIs was undertaken by using the number of ball possessions, as previously described [1,14]. A ball possession was defined as a period of play between when one team gains control of the ball and when the other team gains control of the ball [15]. According to this definition, Offensive Rebounds are included in the same possession. Ball possessions were calculated by the following equation: ball possessions = (field goals attempted) – (Offensive Rebounds) + (Turnovers) + 0.44 × (Free Throws attempted) [16]. The study was conducted according to the ethical guidelines of the authors’ affiliated institutions but did not require Ethics Committee approval, because a non-interventional design was used, whereby all analyzed data were de-identified and available in the public domain.

### 2.2. Reliability and Validity of Data

To confirm the reliability and accuracy of data, a sub-sample of 10 games was randomly selected and observed by two experienced analysts (i.e., basketball coaches with more than five years of experience in basketball performance analysis). Comparisons between results from the analysts and the FIBA official website produced perfect Intra-Class Correlation Coefficients (ICC = 1.0) for most KPIs (i.e., Paint Score, Mid-Range Score, Three-Point Score, Free Throws, Offensive and Defensive Rebounds, Assists, Blocks, Personal Fouls and Steals). A lower and acceptable ICC (0.88) was obtained for the final KPI, Turnovers. The reason for lower reliability of Turnovers is possibly the result of the differences of video and statistical software.

### 2.3. Statistical Analysis

The effects of KPI on match outcome (final point differential) were examined via MLR and QR models. The MLR analysis was based on the average relationship between a set of independent variables and the dependent variable by the conditional mean function, E(y/x), which provided only a partial view of the relationship [10]. In contrast, the QR analysis described the relationship at different points in the conditional median or quantile distribution of dependent variable, Qq(y/x), where q was the quantile or percentile, and the median was the 50th percentile of the empirical distribution with no zero values for the dependent variable [10]. When interpreting the results for the above models, positive and negative coefficients indicated a greater/lower propensity to increase/decrease match outcome [17]. In order to examine the influence of KPIs on the extreme distribution tails for match outcome, our study selected five quantile levels (0.10, 0.25, 0.50, 0.75 and 0.90) for the QR model. Notably, the quantile levels Q10 and Q25 represented the lower tail distribution of final match outcome, and the quantile levels Q75 and Q90 represented the higher tail distribution of final match outcome. All data analyses and visualization were conducted by using R software (R project version 3.5.1), with statistical significance set at *p* < 0.05. 

## 3. Result

The parameter estimates of the MLR as a baseline model and QR at five selected quantile levels (0.10, 0.25, 0.50, 0.75 and 0.90) are presented in Table 2. The visualization of the results obtained for the advanced QR analyses are shown in Figure 1. The summary of significant KPIs, according to the MLR and QR models, is shown in Figure 2. 

The horizontal axis lists the different quantiles, while the vertical axis values represent the regression coefficient. The black solid line with yellow dots is the estimate of the regression coefficient for each quantile (0.10, 0.25, 0.50, 0.75 and 0.90), while the light orange shaded area represents the 95% confidence intervals of the coefficients. The red continuous line corresponds with the multiple linear regression coefficients, with the deep gray shaded area representing the 95% confidence intervals of the multiple linear regression estimate.

### 3.1. Offensive Variables

A positive relationship between Paint Score and match outcome was identified for MLR. A similar relationship was only evident at the 25th quantile for QR analyses with a one-unit increase in Paint Score, leading to a match outcome increase of 0.206 units. A positive impact for Mid-Range Score on match outcome was identified at the 10th quantile only during QR analysis, with no relationship identified for the MLR analysis (Table 2). Three-Point Score (*p* < 0.01) had a positive effect on match outcome for the MLR analysis and all quantiles of the QR analyses, except for the 90th quantile. Offensive Rebounds were positively associated with match outcome at the 90th quantile for the QR analysis, but no such relationship resulted for the MLR analysis. Turnovers were positively associated with match outcome for MLR and all QR quantiles, except for the 10th quantile (Table 2). No significant relationships between match outcome and Free Throws or Assists were evident for MLR or QR analyses (Table 2).

### 3.2. Defensive Variables

Defensive Rebounds had a negative impact on match outcome for MLR and QR (90th quantile), analyses with a one-unit shift associated with a change of 0.425 and 0.954, respectively. Increasing Defensive Rebounds can reduce final point differential. Steals were positively associated with match outcome for MLR and QR (10th and 25th quantiles) analyses, with a one-unit shift resulting in a change of 0.625, and 0.765 and 0.586, respectively. Personal Fouls were the only KPI negatively associated with match outcome for MLR and all QR quantiles, with a one-unit shift predicting a 0.661, and 0.234–1.735-unit change, respectively. No significant relationships between match outcome and Blocks were evident for MLR or QR analyses (Table 2).

### 3.3. Situational Variables

The quality of opponent (situational variable) had no impact on match outcome via MLR or QR analyses across the entire range of quantiles (Table 2 and Figure 2). 

## 4. Discussion

The aims of this study were to model the relationship between match outcome and KPIs and compare these relationships by using MLR and QR models. Our findings indicated that several offensive (Paint Score, Mid-Range Score, Three-Point Score, Offensive Rebounds and Turnovers) and defensive (Defensive Rebounds, Steals and Personal Fouls) KPIs were associated with match outcome. Furthermore, the QR model provided a greater understanding of the influence of KPI on match outcome than the MLR model. Notably, QR analysis highlighted additional KPIs (Mid-Range Score at the 10th quantile, and Offensive Rebounds at the 90th quantile) that were not identifiable via MLR analyses for match outcome. Our results highlight QR modelling as a potentially superior tool for performance analysts and coaches to design and monitor technical–tactical plans during match-play.

Paint Score (25th quantile), Mid-Range Score (10th quantile) and Steals (10th and 25th quantiles) were associated with the lower distribution end (closer matches) of final-match outcome. Additionally, excellent Three-Point Score was associated with match outcome, with this KPI providing opportunities for a range of offensive actions, including dribble penetration, creation of player activity in the paint and difficult defender decisions (i.e., maintain current assignments or help others) [6]. It is worth noting that Three-Point Score may not obtain higher chances of winning during FIBA World Cup, so coaches should focus on improving other aspects (i.e., paint points) for ultimate success, especially for 75th and 90th quantile. These KPIs may also typically lead to more physical contact and a reliance on players to combine strength and their athleticism with skill [4]. For example, speed, agility and repeated sprint ability, or this combination, have previously been reported as fundamental capacities for successful basketball athletes [18] and highlight the need for athletic and versatile players for success at the highest competition level. Moreover, Steals were reported to differentiate men’s teams within close (1 to 12 point differential) matches during the basketball World Championships in 1999–2002 [19]. Previously, Steals were associated with superior speed, agility, anaerobic power and repeated-sprint-ability in male basketball athletes [7]. The inclusion of athletes that possess these specific fitness characteristics for national teams may provide the talent base to enhance this KPI during match-play and lead to ultimate success [8]. Additionally, development of these fitness characteristics, specific to each playing position, during the pre-World Cup period, may be suggested as a priority for coaches in their preparation for the competition [8]. Concurrently, more detailed scouting feedback about opponents’ technical characteristics could help coaches to better prepare match strategies. Importantly, our study highlighted that scoring and other KPIs representative of aggressive defensive behaviors (Steals and Personal Fouls) were vital to success in the FIBA Basketball World Cup [20,21]. Personal Fouls were the only KPI negatively associated with match outcome across the entire range of quantiles. This relationship was prominent, especially for disparate matches, with estimates increasing from the lower to the higher end of the distribution of match outcome. This trend was notable, as committing fouls will allow opponents access to an easy scoring opportunity (i.e., Free Throws), reduce match pace and disrupt fixed tactical strategies [22]. Such actions will make it tougher for opponents to execute open-court plays [23] and force the opponent to lose ball possession (Turnovers) for greater match success [2,24]. In fact, most Turnovers are created by offensive bad decisions and/or defensive good decisions, depending on the given situation [8]. The incorporation of decision-making tasks during training and awareness of specific situations that may create Turnovers [25,26] will assist players and coaches for competition. Collectively, these results support the development of shooting proficiency, key defensive strategies by coaches, selection of athletes highly proficient in defensive actions and development of players’ decision-making capacity as essential for potential success of national teams.

As indicated above, both offensive and defensive KPIs were associated with match success. Similarly, Offensive and Defensive Rebounds (90th quantile) were associated with the upper distribution of match outcome. Our findings were in line with previous studies [8,27,28,29] that identified “defensive rebounds” as the predominant performance indicator to discriminate winning and losing within Olympic basketball games. Similarly, Zhang [30] identified “defensive rebounds” as the key discriminator for success in the NBA. This KPI represents the teams’ ability to recover the ball after an opponent’s missed shots and more opportunities to score points and win the match [31]. Indirectly, this KPI may also reflect high-level performances associated with (i) game pace, with more Defensive Rebounds indicating more fast-break ball possessions; and (ii) players somatic characteristics, with taller and stronger players securing more rebounds, likely due to superior stretch-shortening-cycle jumping performances [17,18]. Therefore, selection of players with these characteristics, and incorporation of a faster playing style, may provide greater match success at the elite international level.

One of the most important findings of the current study was that the quality of opponent had no impact on match outcome. This was unexpected, as opponent quality has been reported to be influential for match success within the Spanish Basketball Professional League and National Basketball Association [4,32]. Potentially, team quality between teams was similar because all national teams undergo an elimination process to qualify for the FIBA Basketball World Cup [33]. 

This is the first study that has identified the KPIs associated with match outcome, using QR analysis. The QR analyses enabled an examination of technical differences at different levels of the distribution for final match outcome. The results indicated that the correlates had dissimilar associations across the entire range of distribution for match outcome, suggesting that the MLR analysis provided a limited mean response for the given level of predictor variables [34]. This average effect may either underestimate or overestimate the effect of covariates at different quantiles of the distribution for match outcome [10]. Thus, QR analysis should be considered as a significant enhancement and superior tool for use in sport sciences. Future applications of QR analyses will confirm its ability to routinely support performance analysts and coaches for greater success in elite basketball and other sports.

While the current study provided novel findings, some limitations need to be addressed in future research. First, the overall sample was relatively modest, despite being from an elite competition. Future research could expand the sample with a longitude design to explore the influence of KPIs on match outcome, across several FIBA Basketball World Cup competitions. Second, the current study only considered the contextual effect of opponent quality on match outcome. Future studies are recommended to examine the interaction effects of other situational variables (e.g., competition stage) on match outcome. Finally, our analyses focused on a male senior competition only, with future studies encouraged to examine other competitions (e.g., female, juniors, etc.). Such work will likely extend upon the current results and applicability of QR analyses, to model technical performances in elite basketball competition. 

## 5. Conclusions

In summary, Paint Score (25th quantile), Mid-Range Score (10th quantile) and Steals (10th and 25th quantiles) were associated with the lower distribution of match outcome, while Offensive and Defensive Rebounds (90th quantile) were associated with the upper distribution for match outcome. Three-Point Score and aggressive defensive KPI (Steals and Personal Fouls) may provide an advantage to winning matches during the FIBA World Basketball Cup. The use of QR modelling can provide sport scientists and coaches with superior and practical approaches to exploring multivariate datasets in basketball science. 

## Figures and Tables

**Figure 1 ijerph-17-05722-f001:**
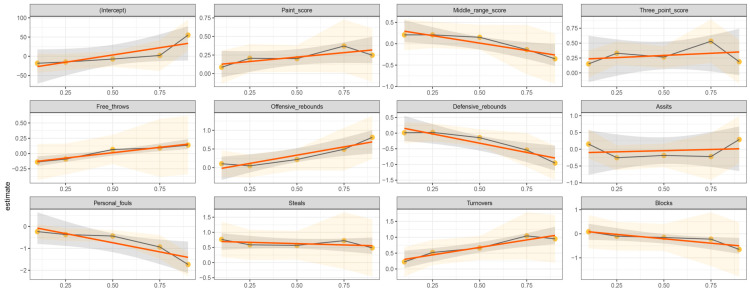
Regression coefficients of MLR and QR for the effect of key performance indicators on match outcome.

**Figure 2 ijerph-17-05722-f002:**
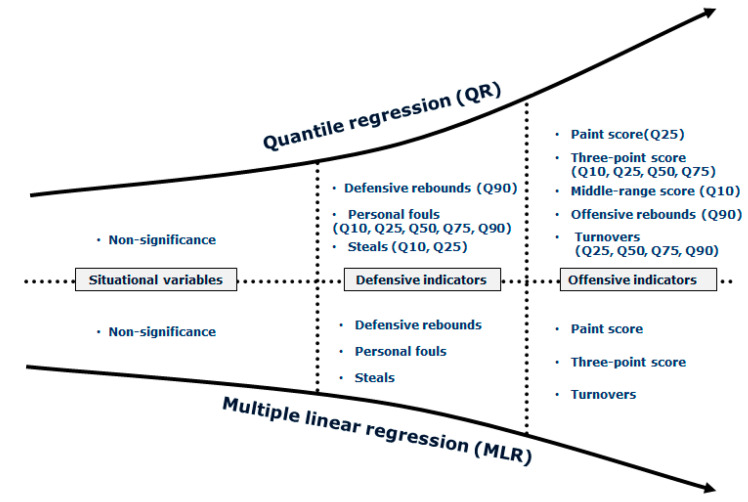
Summary of significant key performances indicators, according to the multiple linear regression (MLR) and the quantile regression (QR) models.

**Table 1 ijerph-17-05722-t001:** The variables selected in the current study.

**Paint Score:** The number of points scored by a player or team in the keyway, also known as the paint area.
**Mid-Range Score:** The number of points scored by a player or team outside of the paint area but inside the three-point line.
**Three-Point Score:** The number of three-point field-goals that a player or team scored.
**Free Throws:** The number of Free Throws that a player or team scored.
**Offensive Rebounds:** The number of rebounds a player or team collected while on offence. **Assists:** An assist occurs when a player completes a pass to a teammate that directly leads to a field goal score.
**Turnovers:** A Turnover occurs when the player or team on offence loses the ball to the defense.
**Defensive Rebounds:** The number of rebounds a player or team collected while on defense.
**Personal Fouls:** The total number of fouls that a player or team committed.
**Steals:** A steal occurs when a defensive player takes the ball away from a player on offence.
**Blocks:** A block occurs when the defense player tips the ball and prevents an offensive player’s shot from scoring
**Quality of opponent:** Strong and weak teams.

**Table 2 ijerph-17-05722-t002:** Parameter estimates from multiple linear regression (MLR) and quantile regression (QR) on final score difference quantiles.

Variables	MLR	Quantile Regression (QR)				
		Q10	Q25	Q50	Q75	Q90
		FPD = 2	FPD = 6	FPD = 16	FPD = 20	FPD = 39
Constant	5.437 (13.031)	−18.186 ** (8.060)	−14.983 (9.872)	−7.236 (12.168)	2.087 (22.027)	55.062 ** (27.521)
Paint Score	0.260 ** (0.116)	0.086 (0.079)	0.206 ** (0.085)	0.201(0.107)	0.371(0.207)	0.248 (0.183)
Mid-Range Score	−0.020 (0.187)	0.207 ** (0.094)	0.207 (0.107)	0.151 (0.179)	−0.139 (0.302)	−0.352 (0.342)
Three-Point Score	0.319 *** (0.118)	0.151 ** (0.074)	0.330 *** (0.083)	0.266 ** (0.134)	0.531 *** (0.172)	0.187 (0.219)
Free Throws	−0.020 (0.154)	−0.138 (0.095)	−0.091 (0.125)	0.066 (0.158)	0.092 (0.270)	0.138 (0.259)
Offensive Rebounds	0.165 (0.185)	0.104 (0.123)	0.048 (0.142)	0.218 (0.185)	0.493 (0.334)	0.811 ** (0.405)
Assists	−0.110 (0.227)	0.154 (0.130)	−0.253 (0.156)	−0.185 (0.250)	−0.218 (0.339)	0.289 (0.414)
Turnovers	0.795 *** (0.241)	0.225 (0.149)	0.521 *** (0.174)	0.661 *** (0.207)	1.043 *** (0.337)	0.950 ***(0.365)
Defensive Rebounds	−0.425 ** (0.174)	0.012 (0.106)	0.023 (0.126)	−0.145 (0.155)	−0.541(0.307)	−0.954 ** (0.369)
Personal Fouls	−0.661 *** (0.176)	−0.234 ** (0.109)	−0.368 *** (0.137)	−0.436 ** (0.176)	−0.937 *** (0.303)	−1.735 *** (0.326)
Steals	0.625 ** (0.304)	0.765 *** (0.184)	0.586 *** (0.190)	0.568 (0.289)	0.723 (0.444)	0.486 (0.529)
Blocks	−0.268 (0.361)	0.086 (0.200)	−0.099 (0.216)	−0.154 (0.299)	−0.223 (0.529)	−0.649 (0.509)
Quality of Opponent	2.400 (2.170)	2.108 (1.376)	1.943 (1.615)	1.800 (2.020)	6.149 (4.188)	3.972 (3.309)

Standard errors in parentheses; FPD = final point differential;** *p* < 0.05 and *** *p* < 0.01.

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
