# Peer review of "Modelling the Relationship between Match Outcome and Match Performances during the 2019 FIBA Basketball World Cup: A Quantile Regression Analysis"

_ijerph, 2020, doi:10.3390/ijerph17165722_

Round 1

Reviewer 1 Report

The work is pretty good. You need to improve the conclusions. Improve key points to highlight at work.

Reviewer 2 Report

Keywords

Line 34: It is possible to add a keyword related to basketball

Materials and methods

Sample:

I would describe on one hand the sample and on the other hand the different variables. The different variables in another section.

Table 1. Specified in the table corresponds to each variable group

Reliability and validity data:

Line 101: add "and" in phrase "(...personal fouls and steals)"

Line 102: add "and" in phrase "...final KPI and turnovers."

Results

Line 145: Note Table 2. p<0.001 change for p<0.01

Discussion

Line 162-163: add "and" in phrase "...10th quantile and offensive rebound..."

Line 213: Explain some more reason about the influence of the opponent's quality.

Conclusions

Line 243: Why is QR better? Specify some more reason.

Reviewer 3 Report

Congratulations to the authors on another article in which they try to quantitatively explain the difference between the winners and the losers teams in major basketball competitions, and thus in basketball in general.

Reviewer 4 Report

First of all, I want to thank to the authors their effort to broaden an interesting topic. The present research has high methodological standards while presenting interesting results and some applicability. I want to highlight the effort to provide results with more robust methods (QR) that can lead to some different interpretations. Moreover, some discrete variables (i.e. points) have been traditionally analyzed as they were continuous which is an important methodological issue. With that being said I see some concerns that need attention.  

Abstract

Line 18: KPI do not improve odds of winning, but helps us to better understand the game.

Introduction

Line 44: not all technical behaviors have been studied/related to match outcome, only some.

Materials and Methods

Why this performance indicators have been chosen over others (i.e. overall 3p shots)?

Line 88: how ‘k’ value for the k-means method have not been detailed.

Line 89: how possession is understood should be described i.e. when there is an offensive rebound, there is a new possession or not?

Table 1: negative variables (fouls, turnovers) should be ordered inversely to adapt their results to the other outcomes increase as good and decrease as bad.

Table 2: values within the parentheses should be described.

Outliers detection would add a beautiful insight to the understanding of highly variables dataset such box scores are.

Results

Figure 1: the MLR method provides the baseline and can have his own CI that would help to understand the differences between both methods. Win and Loss can be established in the y-axis.

Figure 2 is a great and comprehensive overview of the results chapter.

Ranges of every value can provide to the reader (principally to those non familiarized with elite basketball) a better understanding of every variable.

Discussion

Line 166: authors state that some variables are associated with the lower distribution of final match outcome but this comparison has not been established in the methods chapter.

Line 187: there are more variables negatively associated to match outcome at least somewhere in the spectrum.

Line 203: In line 134 the authors presented that defensive rebounds had a negative impact while in the present line are presented as the best indicator, this should be clarified.

Lines 207-210: a defensive rebound is not only a presumable opportunity for a fast-break, but the result of a missed shot of the opponent reflecting the inability to change the scoreboard.

Line 213: The quality of the opponent could affect the result but maybe has not been detected through the study of the present variables.

The lower reliability for turnovers should be explained or discussed.

The threshold values (values above or below which there is no added benefit) i.e. Q75 in the case of three points shots should be discussed and compared, and are of great interest to coaching. Let us say that over fifteen 3p made you do not obtain higher chances of winning, then you can focus on improving other aspects of your team performance i.e. paint points.

‘U’ shaped regressions are of great interest and should be discussed because sometimes indicate that mid-range values are optimal compared to extreme values  (3p, assists).

References

The references are well chosen and provide a good insight on the topic. I rather like to suggest the authors the following to be considered:

García, J., Ibáñez, S. J., De Santos, R. M., Leite, N., & Sampaio, J. (2013). Identifying basketball performance indicators in regular season and playoff games. Journal of human kinetics36(1), 161-168.

Malarranha, J., Figueira, B., Leite, N., & Sampaio, J. (2013). Dynamic modeling of performance in basketball. International Journal of Performance Analysis in Sport13(2), 377-387.
